# Transperineal Laser Ablation of the Prostate (TPLA) for Lower Urinary Tract Symptoms Due to Benign Prostatic Obstruction

**DOI:** 10.3390/jcm12030793

**Published:** 2023-01-19

**Authors:** Francesco Sessa, Paolo Polverino, Giampaolo Siena, Claudio Bisegna, Mattia Lo Re, Pietro Spatafora, Alessio Pecoraro, Anna Rivetti, Luisa Moscardi, Marco Saladino, Andrea Cocci, Mauro Gacci, Vincenzo Li Marzi, Marco Carini, Andrea Minervini, Riccardo Campi, Sergio Serni

**Affiliations:** 1Unit of Urological Robotic Surgery and Renal Transplantation, University of Florence, Careggi Hospital, 50100 Florence, Italy; 2Department of Experimental and Clinical Medicine, University of Florence, 50100 Florence, Italy; 3Unit of Oncologic Minimally Invasive Urology and Andrology, University of Florence, Careggi Hospital, 50100 Florence, Italy

**Keywords:** transperineal laser ablation of prostate (TPLA), benign prostatic hyperplasia (BPH), lower urinary tract symptoms (LUTS), ultra-minimally invasive surgical techniques (uMISTs), ejaculation sparing

## Abstract

We aimed to review the current evidence on surgical and functional outcomes of Transperineal Laser Ablation for LUTS due to BPH. A comprehensive review of the English-language literature was performed using the MEDLINE and Web of Science databases until 1 August 2022, aiming to select studies evaluating TPLA for the treatment of LUTS due to BPH. Additional records were found from Google Scholar. Data were extracted and summarized in Tables. An appropriate form was used for qualitative data synthesis. Seven studies were included in the review, with all being single arm, non-comparative studies. In all studies, functional outcomes were evaluated with uroflowmetry parameters and validated questionnaires, showing a promising effectiveness at short- and mid-term follow-up. There is a lack of standardized pathways for preoperative assessment of patients suitable for TPLA, and even the technique itself has been reported with a few nuances. A good safety profile has been reported by all the authors. Although promising results have been reported by different groups, selection criteria for TPLA and few technical nuances regarding the procedure were found to be heterogeneous across the published series that should be standardized in the future. Further research is needed to confirm these findings.

## 1. Introduction

Benign Prostatic Hyperplasia (BPH) is one of the most common causes of lower urinary tract symptoms (LUTS) in adult men resulting in a major impact on the quality of life (QoL). The current guidelines recommend change in the lifestyle and pharmacological therapy as first line of treatment, followed by surgery in case of failure or intolerance [1].

Nevertheless, the pharmacological treatment has a considerable impact on the sexual sphere, in particular on the ejaculatory function, leading many patients to have low adherence rates or to discontinue the therapy [2,3]. On the other hand, surgical techniques are not devoid of side effects (rate of ejaculatory disfunction ranges from 11 to 70%) and require general or spinal anaesthesia and hospitalization [4,5].

During the last two decades, surgical treatment for BPH has evolved from open techniques to endoscopic techniques, including the use of laser technology, to ultra-minimally invasive surgical techniques (uMISTs) including, Aquablation, Urolift, Rezūm, temporary implantable nitinol device (TIND) and prostatic artery embolization (PAE). The aim is to effectively treat symptoms, while having as few side effects as possible, and a rapid recovery [1,6,7,8]. In this challenging scenario, Transperineal Laser Ablation (TPLA) has been proposed as an alternative ejaculation-sparing management for patients with BPH.

To date, preliminary studies have shown favorable perioperative and functional outcomes after TPLA for carefully selected patients with LUTS due to BPH. However, there is still no consensus regarding the treatment protocol, or the best indications for TPLA, and although a number of systematic reviews were previously conducted on surgical and functional outcomes of other uMISTs, to the best of our knowledge no review has specifically been focused on TPLA, yet [8,9,10].

To fill this gap, we provide a comprehensive overview of the current role of TPLA in clinical practice, focusing on surgical indications, perioperative outcomes, efficacy and adverse events.

## 2. Materials and Methods

A comprehensive review of the English-language literature was performed using the MEDLINE (via PubMed) and Web of Science (WOS) databases until 1 August 2022, using a combination of free text and MeSH subject headings, aiming to select prospective or retrospective studies evaluating the outcomes of TPLA for the treatment of patients with LUTS due to BPH. Additional records were found from the references of the retrieved studies, as well as manual searches in Google Scholar. A separate search was conducted using www.clinicaltrials.gov to identify current clinical trials evaluating TPLA for patients with BPH. A flowchart depicting the overall review process according to the PRISMA statement has been reported in Appendix A [11]. Data from the studies included in the review were extracted using a priori-developed data extraction forms and are summarized in five Tables (Table 1, Table 2, Table 3, Table 4 and Table 5). Qualitative data synthesis owing to the heterogeneity of the studies was assessed, while a quantitative synthesis was not performed.

Risk-of-bias was assessed independently by two authors (M.L.R. and P.P.) according to the Quality In Prognosis Studies (QUIPS) tool, as shown in Appendix A. This tool provides a measure of the risk of bias and applicability over six domains of interest. Once again, in case of disagreement, a third party was involved to solve the issue. The proportion of studies with a low risk of bias in study participation, attrition, prognostic factor measurement, outcome measurement, study confounding, and statistical analyses and reporting domains was 86%, 86%, not available, 43%, 0% and 0%, respectively.

## 3. Results

A graphical overview of the main review findings is reported in Figure 1. The figure summarizes device characteristics, setting (outpatient clinic **vs.** operating room; day-hospital vs. hospitalization; anaesthesia), indications in terms of prostate volume, type of operator and outcomes of the studies included, in order to provide an overview of the available evidence on these topics and to stress the differences among studies in this clinical scenario.

### 3.1. Surgical Technique: An Overview

TPLA is an ultrasound (US) guided minimally invasive procedure requiring a biplanar TRUS and EchoLaser™ system consisting of a multisource diode laser with four independent laser sources, operating at 1064 nm wavelength (EchoLaser, El.En. S.p.A, Calenzano, Italy) and a dedicated planning tool (ESI—Echolaser Smart Interface, Elesta S.p.A, Calenzano, Italy) with simulation software that allows the user to plan the treatment and to place applicators in the prostate in a safe manner [11,12,13,14,15,16,17]. This EchoLaser application is also known as SoracteLite™ (El.En. S.p.A, Calenzano, Italy). A catheter placement and local anesthesia are needed before starting the procedure. The laser light is conveyed by the source to the tissues through 300 µm quartz optical fibers with a flat tip (Fiber Optic for PLA, Elesta S.p.A., Calenzano, Italy), which are inserted percutaneously through 21 G Chiba needles (Introducer, Elesta S.p.A., Calenzano, Italy) under transrectal ultrasound guidance. The laser light produces an ellipsoidal shape area of coagulative necrosis around the tip of the fiber (approximately 2/3 extended beyond the fiber tip and 1/3 behind it depending on the power and dose applied). A needle placement verification is required to guarantee the right safety distances from the urethra and from the bladder neck. The procedure can be planned via the Echolaser Smart Interface (ESI), a dedicated device that allows the operator to establish the correct ellipsoidal shape area of coagulative necrosis on the prostatic tissue [12,14,15,16,17]. Once the fibers are placed, the energy can be delivered. The laser causes hyperthermia, denaturation and coagulative necrosis of proteins. The maximum volume treated in a session and the extent of the ablation vary according to the prostatic volume, anatomy and surgeon preference. In some cases, especially in larger prostates, a pull back of applicators (retraction of 5–10 mm along its trajectory) during the same treatment session allows for the ablation of another part of the prostatic tissue not treated in the previous illumination; delivering additional laser energy.

### 3.2. Characteristics of the Included Studies and Patients’ Characteristics

The key characteristics of the studies contained in the review are reported in Table 1. Overall, seven studies from different countries (six from Italy, one from China) conducted between 2014 and 2021 were included in our review [12,13,14,15,16,17,18]. Of these, five studies were prospective and monocentric, while two studies were retrospective [13,17] (one of which multicentric) [17]. In particular, all the studies were single arm, non-comparative studies. The number of patients across the included studies ranged between 18 and 160. Preoperative patient characteristics are depicted in Table 1. The mean age ranged from 61.9 to 73.9 years. Three studies (43%) reported patients’ median BMI and Charlson Comorbidity Index [12,14,15]. All patients experienced mild to severe urinary symptoms according to an IPSS questionnaire, with a non-negligible impact on their quality of life. Voiding measurements such as Qmax and PVR were recorded in all the studies, showing a similar baseline status between the populations, with the exception of quite higher PVR values in Patelli’s series (mean PVR 199.9 ± 147.3 mL) [18]. Only a proportion of studies provided information on patients’ sexual function at enrolment, in particular four studies (57%) [12,14,15,16] reported preoperative IIEF-5 questionnaire results, and only three [12,14,16] and one [14] studies reported preoperative MSHQ-EjD-3 items and MSHQ-EjD-bother, respectively.

### 3.3. Indications for EchoLaser TPLA

Inclusion and exclusion criteria were heterogeneous among the studies, as can be noticed from the variability of the median prostate volume in the included studies, which ranged from 40 to 102 mL. BPH therapy status of patients has been described in four studies [12,14,15,16]. Patients taking different medications for BPH were included [patients not taking drugs, patients on alfalitic therapy, and patients on combination therapy (alfalitic + 5ARi)]; one study analyzed only patients on combination therapy [16]. A lack of homogeneity in patient selection was also seen in the choice of treating patients with an indwelling catheter, included by the authors in their analysis of four out of seven studies [12,15,17,18]. The presence of a middle lobe larger than 10 mm was considered a contraindication only in one study [16].Other studies included in this review showed that the treatment of patients with a middle lobe was feasible in their experience [14,15,17,18]. Although all the authors, except Cai and Patelli [13,18] declared that antiplatelet or anticoagulant therapy was not a contraindication, only in one paper were the number of patients undergoing antiplatelet and/or anticoagulant therapy at the time of surgery reported [12].

### 3.4. Intraoperative Features

A detailed overview of operative features is shown in Table 2.

The procedure was performed under lidocaine 2% local anaesthesia in all the cases, with concomitant conscious sedation in five studies [12,14,15,17,18]. For all procedures the same diode laser generator, that utilizes four independent channels for simultaneous firing using Elesta was used. Even if the laser system was the same, different settings were adopted. In fact, Sessa, De Rienzo and Manenti started with a higher power setting (5 W, 4.5 W and 5 W, respectively) [12,14,16], reducing the power after a few minutes, in contrast to a 3 W fixed setting adopted by other authors. Energy setting of the single fiber was 1800 J in almost all cases, with the exception of Patelli and Sessa who reported different settings ranging between 1200 and 1800 J [12,18]. Up to three fibers per lobe were used with simultaneous laser emission, depending on prostate dimensions and surgeon preference. A contrast-enhanced ultrasound or MRI measuring the coagulation zone volume was performed postoperatively in three [13,17,18] and one [16] studies respectively, but only Cai and Manenti reported their results [13,16]. The procedure required a brief hospitalization even when it was conducted in an outpatient setting in two studies [12,16]. All the authors adopted the pull-back, but only Frego reported precise data about the technique, in particular it was used in the case of prostate volume >80 mL (in 31.8% of cases) [15].

### 3.5. Functional Outcomes

Table 3 reports perioperative data of the studies included in this review. The catheterization time varied according to the individual treatment protocol. Pacella et al. [17], in the absence of adverse events, removed the catheter at the end of the treatment, while in four studies a standard catheter removal on the seventh postoperative day was performed, in the absence of complications [12,14,15,16]. Overall, none of the authors reported catheterization times longer than a month. Similarly, there is variability among studies about postoperative therapy and in some cases no data were reported in this regard [13,17,18]. However, a common choice was to administer antibiotics and anti-inflammatory drugs for at least 5–7 days, and in some cases to continue BPH therapy for one month [12,14,15,16]. Notably, follow-up data were almost all short and mid-term data, with different timings and modalities; only De Rienzo reported a median follow-up of 16 months; however, the data at 12 months and onwards were not recorded via validated questionnaires or diagnostic exams but with a follow-up performed via a telephone interview [14].

In all studies, in the controls at 1,3, 6 and 12 months, depending on the different follow-up protocols, there was a significant but durable improvement in all the functional outcomes examined, as shown in Table 4 and 5. Four authors reported postoperative IIEF-5 questionnaire results and no impact on erectile function was observed in any case [12,14,15,16]. Ejaculatory function was preserved in almost all cases, or even in some cases there was an improvement in MSHQ score [12,14,16], while 2/160 and 1/22 patients in Pacella and Manenti series, respectively, reported loss of ejaculatory function, although not measured by a questionnaire [16,17]. A reduction in prostatic volume after treatment was also observed [13,15,16,17,18].

### 3.6. Intra and Postoperative Complications

Overall, complication rate ranged from 0 to 13% [12,13,14,15,16,17,18]. A single case of urethral burn was the only intraoperative complication reported [13]. There were mostly early (<30 days) complications, only one case of urinary tract infection (UTI) occurred after 45 days from the procedure [15]. The most common complications were urinary retention and UTIs and they were treated with re-catheterization and antibiotics, respectively. Three cases of prostatic abscess (in one case bilateral) requiring percutaneous drainage (Clavien-Dindo III) were recorded [14,15,16]. Interestingly, Frego et al. reported a significantly higher complication rate when compared to other studies [15]. However, it might be related to the fact that dysuria was included among the complications, not considered as such by other authors, and it alone constituted 61% of the total complications reported. A more accurate description of complications and their treatment are displayed in Table 5.

## 4. Discussion

There is an increasing interest among urologists for ejaculation-sparing options for the management of patients with LUTS due to BPH [19,20,21,22]. In this clinical scenario, EchoLaser TPLA has been introduced as a novel minimally-invasive procedure with favorable results across different settings [12,13,14,15,16,17,18,23]. Nonetheless, several systematic reviews are available for other techniques but, to the best of our knowledge, no previous literature review was specifically focused on TPLA [8,9,10]. Our paper reports an overview of the current management of patients treated with TPLA for BPH—including patient selection, perioperative and postoperative outcomes and complications—providing several insights for both clinicians and patients.

A first key finding of our review is that EchoLaser TPLA showed promising results in terms of functional outcomes and patient safety. Yet, it should be noted that the included studies showed heterogeneous practice patterns regarding postoperative management. In particular, in the early-postoperative setting, described only in four out of seven studies, different therapeutic schemes were adopted by the authors [12,14,15,16]. In all four papers anti-inflammatory drugs and antibiotics were prescribed, but of different classes (NSAIDs and corticosteroids, cephalosporins and fluoroquinolones) and with different dosages. This is a topic for which further research is needed, given the possible implications, especially in terms of infectious complications. Similarly, the different timing of catheter removal could be critical to prevent storage LUTS after surgery or even post-operative urinary tract infections. Furthermore, although all the authors adopted almost the same parameters for the evaluation of the flowmetry outcomes (uroflowmetry parameters and validated questionnaires), the evaluation of the sexual outcomes and the follow-up schedules were remarkably heterogeneous. Specifically, ejaculatory function was evaluated with MSHQ-EjD 3 items and MSHQ-EjD bother questionnaires only in three [12,14,16] and one study [14] respectively. Of note, using questionnaires about ejaculatory function might provide a more comprehensive evaluation of patients, since TPLA might be offered to selected patients who wish to avoid side effects on ejaculation given by surgery or alpha-blocker drugs. As such, reporting the preoperative BPH therapy status of patients, described only in four studies, is also important, both in patient selection and postoperative follow-up [12,14,15,16].

From a surgical perspective, a second key finding of our review is that the technique itself, although similar across studies, has been reported with a few nuances in terms of energy and number of fibers used; factors mainly affected by the prostatic volume. An example that can reinforce this concept is the use of the “pull-back” technique, adopted in all studies to widen the treatment area in large prostates. However, only one author established a cut-off in terms of prostate volume above which the pull-back was performed [15]. Nonetheless, within this context, the surgeon learning curve can significantly affect the results, as highlighted by Manenti and Pacella [16,17]. In particular, the complication rate and the procedural time are expected to be lowered by an increase in the experience of operators, with a corresponding improvement in the functional outcomes.

A third key finding of our review pertains to the lack of standardized pathways for the preoperative assessment of patients who are candidates for TPLA. This is important as indications for EchoLaser TPLA treatment according to the current literature were highly heterogeneous (Table 1). Prostate volume was extremely heterogeneous among the studies, ranging from 40 mL in the De Rienzo experience [14], to over 100 mL in Manenti’s work [16], pointing out a lack of a clear consensus on the ideal prostate volume liable to be treated with TPLA. Of note, Manenti et al. excluded patients from their study who had a large middle lobe (>10 mm) [16]. A lack of homogeneity in patient selection can also be found regarding patients with an indwelling catheter, included in only four out of seven studies [12,15,17,18]. Another key point, potentially related to the risk of postoperative adverse events, is the management of patients on anticoagulant or antiplatelet therapy. In fact, with a few exceptions [12], the proportion of patients on anticoagulant/antiaggregant therapy undergoing TPLA was not reported in most reports.

A limitation at a study-level is the lack of long-term follow-up and comparative randomized controlled trials (RCT) in this setting, which hinder clinically meaningful conclusions on the safety and efficacy of EchoLaser TPLA beyond the short-term. This aspect is certainly attributable to the novelty of the technique. Yet, more research is needed to evaluate the rate of pharmacological or surgical re-intervention in the long term after EchoLaser TPLA, as well as the feasibility and safety of endoscopic re-treatments for TPLA failures. Our review could not find any published RCT or prospective study evaluating the comparative effectiveness of TPLA as compared to either gold-standard endoscopic techniques (such as TURP or Holep) or novel uMISTs. Nevertheless, four trials are ongoing, including an RCT comparing TPLA to transurethral resection of the prostate (TURP) which should provide the first results by the end of 2022 [24,25,26,27]. Additional research projects are also ongoing in this area [28,29]. Our review might provide several key findings to the current scenario. In particular, while the role and indications for TPLA and other ultra-minimal invasive therapies for LUTS due to BPO are still controversial, from a guidelines perspective, it is important for clinicians and patients to have an overview of the available evidence on these topics. In this regard, our review has summarized the available studies on TPLA focusing on its indications, settings and outcomes, providing a foundation to improve shared decision-making in clinical practice. In addition, despite the recent spread of TPLA worldwide, this review underlines the lack of standardization regarding surgical indications, different technical approaches and postoperative outcomes.

Our review is not devoid of limitations. We could not perform any quantitative synthesis of the literature due to the relatively low number and heterogeneity of the included studies. At a study-level, most studies were limited by the small sample size, heterogeneous selection criteria, short follow-up and lack of standardized endpoints. As such, our findings should be interpreted with caution, especially regarding the mid- and long-term safety and functional results of TPLA.

## 5. Conclusions

This is the first comprehensive review providing evidence on the safety and efficacy of TPLA for patients with LUTS due to BPH.

Overall, promising intra-, perioperative, and functional results have been reported by different groups. EchoLaser TPLA has indeed been shown to have a good safety profile and to achieve favorable short-term functional outcomes as well as sexual outcomes. Yet, selection criteria for EchoLaser TPLA, including the ideal patient- and prostate-related characteristics, and few technical nuances regarding the procedure, were found to be heterogeneous across the published series and warrant further investigation.

Further research is needed to confirm these findings in a larger prospective report, as well as to compare the results of TPLA to those of established endoscopic techniques and other uMISTs.

## Figures and Tables

**Figure 1 jcm-12-00793-f001:**
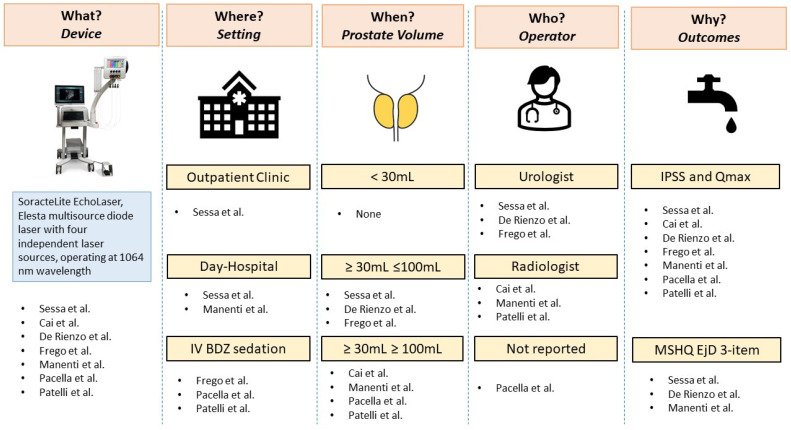
The five Ws of TPLA. IV BDZ: intravenous benzodiazepine; IPSS: International Prostate Symptom Score; Qmax: maximum flow rate; MSHQ EjD: Male Sexual Health Questionnaire Ejaculation Disfunction [12,13,14,15,16,17,18].

**Table 1 jcm-12-00793-t001:** Key characteristics of studies on Transperineal Laser Ablation (TPLA) of prostate included in this review and patients’ characteristics.

Study	Study Type and Design	Year Country	N. Patients	Age (Years)	BMI (kg/m^2^)	CCI	Preop PSA (ng/mL)	Preop IPSS	Preop QoL	Preop Qmax (mL/s)	Preop PVR (mL)	Preop PV (mL)	Preop IIEF-5	Preop MSHQ- EjD 3 Items	Preop MSHQ- EjD Bother	Preop BPH Therapy N (%)	Median Lobe	Indwelling Catheter	Antiaggregant/ Anticoagulant Therapy
Sessa et al. [12]	Non comparative; Prospective; Monocentric	2021; Italy	30	72 (64–79)	28 (24–31)	2 (1–2)	1.64 (0.56–2.43)	21.5 (18.0–27.8)	4.0 (4.0–5.0)	9.5 (7.6–11.2)	100 (70–150)	42 (40–53)	16.0 (7.5–23.5)	5.0 (3.0–7.4)	not reported	α -blockers 16 (53.3); 5-ARI 6 (20.0); Combined therapy 4 (13.3)	Not reported	Included	11 (36.7)
Cai et al. [13]	Non comparative; Retrospectivee; Monocentric	2018–2020; China	20	73.9 ± 9.2	not reported	not reported	not reported	22.7 ± 5.3	4.9 ± 1.7	8.5 ± 3.0	78.7 ± 58.8	70.8 ± 23.8	not reported	not reported	not reported	not reported	Not reported	Not reported	not reported
De Rienzo et al. [14]	Non comparative;Prospective; Monocentric	2018–2019; Italy	21	62 (54–69)	27 (25–28)	2 (1–2)	2.0 (1.33–3.0)	18 ± 3.9	4.1 ± 1.0	9.2 ± 3.4	81.8 ± 62.6	40 (40–50)	17.8 ± 6.6	5.7 ± 4.5	1.2 ± 0.5	α -blockers 14 (66.7); 5-ARI 10 (47.6); Combined therapy 8 (38.1)	Included	Excluded	not reported
Frego et al. [15]	Non comparative;Prospective; Monocentric	2019–2020; Italy	22	61.9 (55–65.5)	27.16 (24.8–28.6)	1 (1–2)	2.24 (1.4–4.5)	22 (19.5–25.25)	4 (4–5)	9 (5–12.5)	60 (25–107.5)	65 (46.5–81)	22 (16.5–24)	not reported	not reported	α-blockers 22 (100%); Combined therapy 6 (27.3%)	Included	Included	not reported
Manenti et al. [16]	Non comparative;Prospective; Monocentric	2018–2020; Italy	44	72.1 ± 6.6	not reported	not reported	7.3 ± 1.8	18.5 ± 5.5	5.8 ± 1.4	7.6 ± 4.2	138.4 ± 40.8	102.42 ± 36.3	21 ± 4	4.9 ± 3.7	not reported	Combined therapy 44 (100%)	Excluded [if dimension superior of 10 mm]	Excluded	not reported
Pacella et al. [17]	Non comparative;Retrospective; Multicentric	Not reported; Italy	160	69.8 ± 9.6	not reported	not reported	not reported	22.5 ± 5.1	4.5 ± 1.1	8.0 ± 3.8	89.5 ± 84.6	75.0 ± 32.4	not reported	not reported	not reported	not reported	Included	Included	not reported
Patelli et al. [18]	Non comparative;Prospective; Monocentric	2014–2016; Italy	18	71.7 ± 9.4	not reported	not reported	not reported	21.9 ± 6.2	4.7 ± 0.6	7.6 ± 2.7	199.9 ± 147.3	69.8 ± 39.9	not reported	not reported	not reported	not reported	Included	Included	not reported

Data presented as mean ± SD or as median (interquartile range). BMI: Body Mass Index; CCI: Charlson Comorbidity Index; PSA: Prostate Specific Antigen; IPSS: International Prostate Symptom Score; QoL: Quality of Life; Qmax: Maximum flow rate; PVR: Post Void Residual; PV: Prostate Volume; IIEF-5: International Index of Erectile Function, five items; MSHQ-EjD: Male Sexual Health Questionnaire—Ejaculatory Disfunction; BPH: Benign Prostatic Hypertrophy; 5-ARI: 5-alpha-reductase inhibitors; Combined therapy: 5-alpha-reductase inhibitors + α-blockers.

**Table 2 jcm-12-00793-t002:** Operative and perioperative features and details of technique of studies on Trans Perineal Laser Ablation (TPLA) of prostate included in this review.

Study	Anesthesia	Anesthesia Description	Laser System	Power Setting (W)	Energy Setting (J)	No. of Fibres Per Lobe	Minimum Distance from Bladder Neck (mm)	Minimum Distance from Urethra (mm)	Minimum Distance between Needles (mm)	Procedural Time (min)	Ablation Time (min)	Energy Deployed (J)	Coagulation Zone (mL)	Ablation Range Evaluation	Hospitalization Time (Days)
Sessa et al. [12]	Conscious sedation; local anesthesia	Oral benzodiaze; lidocaine-prilocaine 5% cream; 2% lidocaine	SoracteLite EchoLase, Elesta	5 reduced to 3.5 W after 2 min	1400–1800	1	15	8	Not reported	31.5 (28–37)	Not reported	Not reported	Not reported	Not reported	0
Cai et al. [13]	Local anesthesia	2% lidocaine	SoracteLite EchoLaser, Elesta	3	1800	2	15	8	15	60.9 ± 10.8	42.6 ± 9.9	7179.2 ± 2815.7	11.7 ± 5.8	CEUS	1.5 ± 0.5
De Rienzo et al. [14]	Conscious sedation; local anesthesia	Not reported; 2% lidocaine	SoracteLite EchoLaser, Elesta	4.5 reduced to 3.5 after 1 min	1800	1–2	15	8	10	36.0 ± 9.5	Not reported	Not reported	Not reported	Not reported	0.86 ± 0.15
Frego et al. [15]	Conscious sedation; local anesthesia	Midazola; 2% lidocaine	SoracteLite EchoLase, Elesta	3	1800	1 (54.5%); 2 (45.5%)	Not reported	10	Not reported	Not reported	17.2 (10–18.8)	Not reported	Not reported	Not reported	1
Manenti et al. [16]	Local anesthesia	2% lidocaine	SoracteLite EchoLaser, Elesta	5 reduced to 3 after 2 min	1800	1–2	15	10	8	28.2 ± 10.6	Not reported	Not reported	Not reported	MRI	0
Pacella et al. [17]	Conscious sedation; local anesthesia	Midazola; 2% lidocaine	SoracteLite EchoLaser, Elesta	3	1800	1–3	15	8	8	44.1 ± 12.9	23.4 ± 10.2	6616.2 ± 3880.4	Not reported	CEUS	1.8 ± 0.4
Patelli et al. [18]	Conscious sedation; local anesthesia	Midazola; 2% lidocaine	SoracteLite EchoLaser, Elesta	3	1200–1800	1 (22%); 2 (78%)	15	8	15	43.3 ± 8.7	15.9 ± 3.9	10522 ± 3290.5	10.3 ± 3.6	CEUS	1.5 ± 0.4

Data presented as mean ± SD or as median (interquartile range). CEUS: contrast enhanced ultrasound; MRI: magnetic resonance imaging.

**Table 3 jcm-12-00793-t003:** Perioperative (<30 days) surgical and functional outcomes of studies on Trans Perineal Laser Ablation (TPLA) of prostate included in this review.

Study	Catheterization Time (Days)	Postoperative Therapy	Median Follow-Up (Months)	Time to First Re-Evaluation (Months)	1-mo IPSS	1-mo QoL	1-mo Qmax	1-mo PVR	1-mo PV	1-mo IIEF-5	1-mo MSHQ-EjD 3 Item	1-mo MSHQ-EjD Bother	1-mo PSA (ng/mL)
Sessa et al. [12]	7 (7–8)	Cefixime 400 mg for 7 days; Ibuprofen 600 mg twice a day for 7 days; Gastroprotective therapy for 7 days	3	1	14.5 (12.0–17.8)	3.0 (2.0–3.75)	10.5 (8.0–16-0)	50 (20–100)	Not reported	18.0 (15.0–24.0)	7.5 (4.0–13.1)	Not reported	1.52 (0.93–1.87)
Cai et al. [13]	16.5 ± 4.2	Not reported	6	6	Not reported	Not reported	Not reported	Not reported	Not reported	Not reported	Not reported	Not reported	Not reported
De Rienzo et al. [14]	8.7 ± 2.5	Antibiotics (fluorochinolones or cephalosporine) for 5 days; prednisone 25 mg for 15 days; preoperative BPH therapy for 30 days; bromelain tablets	16	1	12.0 ± 5.6	2.4 ± 1.6	12.1 ± 6.4	37.4 ± 25.7	Not reported	17.4 ± 5.0	9.6 ± 4.1	1.9 ± 1.2	3.0 ± 1.9
Frego et al. [15]	7	Levoxacin for 5 days; dexamethasone 8 mg and ketoprofen 100 mg for 7 days	6 (6–12)	3	Not reported	Not reported	Not reported	Not reported	Not reported	Not reported	Not reported	Not reported	Not reported
Manenti et al. [16]	7	levofloxacin for 5 days; Prednisone 25 mg for 5 days; Alpha blockers for 30 days	12	1	Not reported	Not reported	Not reported	Not reported	Not reported	Not reported	Not reported	Not reported	Not reported
Pacella et al. [17]	11.3 ± 11.5	Not reported	at least 6 mo; 83 patients 12 mo	6	Not reported	Not reported	Not reported	Not reported	Not reported	Not reported	Not reported	Not reported	Not reported
Patelli et al. [18]	17.3 ± 10.0	Not reported	3	3	Not reported	Not reported	Not reported	Not reported	Not reported	Not reported	Not reported	Not reported	Not reported

Data presented as mean ± SD or as median (interquartile range). Mo: months; PSA: Prostate Specific Antigen; IPSS: International Prostate Symptom Score; QoL: Quality of Life; Qmax: Maximum flow rate; PVR: Post Void Residual; PV: Prostate Volume; IIEF-5: International Index of Erectile Function, five items; MSHQ-EjD: Male Sexual Health Questionnaire—Ejaculatory Disfunction.

**Table 4 jcm-12-00793-t004:** Short and mid-term postoperative surgical and functional outcomes of studies on Trans Perineal Laser Ablation (TPLA) of prostate included in this review.

Study	3-mo IPSS	3-mo QoL	3-mo Qmax	3-mo PVR	3-mo PV	3-mo IIEF-5	3-mo MSHQ-EjD 3 item	3-mo MSHQ-EjD Bother	3-mo PSA (ng/mL)	6-mo IPSS	6-mo QoL	6-mo Qmax	6-mo PVR	6-mo PV	6-mo IIEF-5	6-mo MSHQ-EjD 3 item	6-mo MSHQ-EjD Bother	6-mo PSA (ng/mL)
Sessa et al. [12]	13.0 (11.3–6.4)	2.0 (1.75–2.25)	14.2 (11.2–16.3)	40 (25–70)	Not reported	23.0 (17.5–25.0)	8.9 (7.0–16.4)	Not reported	1.51 (0.97–1.79)	Not reported	Not reported	Not reported	Not reported	Not reported	Not reported	Not reported	Not reported	Not reported
Cai et al. [13]	Not reported	Not reported	Not reported	Not reported	Not reported	Not reported	Not reported	Not reported	Not reported	9.1 ± 3.2	2.3 ± 1.3	15.2 ± 4.8	30.3 ± 34.2	54.7 ± 20.9	Not reported	Not reported	Not reported	Not reported
De Rienzo et al. [14]	8.3 ± 3.8	1.4 ± 0.9	13.3 ± 6.7	18.7 ± 21.2	Not reported	17.7 ± 6.7	6.8 ± 3.5	1.3 ± 0.4	1.7 ± 0.8	6.1 ± 2.6	1.7 ± 0.8	13.9 ± 6.2	14.0 ± 16.7	Not reported	18.3 ± 5.7	8.6 ± 3.1	1.4 ± 0.8	1.7 ± 0.8
Frego et al. [15]	8 (4.5–11)	1 (0.5–2)	12 (9–16.5)	39 (10–87.5)	46 (28.4–69)	22 (19.5–24)	Not reported	Not reported	Not reported	5 (3–8.5)	1 (0–2)	15 (11.5–20.5)	40 (16–63)	42.3 (39.5–59)	23 (20.5–24)	Not reported	Not reported	Not reported
Manenti et al. [16]	Not reported	Not reported	Not reported	Not reported	Not reported	Not reported	Not reported	Not reported	Not reported	Not reported	Not reported	Not reported	Not reported	Not reported	Not reported	Not reported	Not reported	Not reported
Pacella et al. [17]	Not reported	Not reported	Not reported	Not reported	Not reported	Not reported	Not reported	Not reported	Not reported	7.7 ± 3.3	1.8 ± 1.0	14.3 ± 3.9	27.2 ± 44.5	60.3 ± 24.5	Not reported	Not reported	Not reported	Not reported
Patelli et al. [18]	10.7 ± 4.7	2.1 ± 1.2	13.3 ± 76.2	81.5 ± 97.8	54.8 ± 29.8	Not reported	Not reported	Not reported	Not reported	Not reported	Not reported	Not reported	Not reported	Not reported	Not reported	Not reported	Not reported	Not reported

Data presented as mean ± SD or as median (interquartile range). Mo: months; IPSS: International Prostate Symptom Score; QoL: Quality of Life; Qmax: Maximum flow rate; PVR: Post Void Residual; PV: Prostate Volume; IIEF-5: International Index of Erectile Function, five items; MSHQ-EjD: Male Sexual Health Questionnaire—Ejaculatory Disfunction; PSA: Prostate Specific Antigen.

**Table 5 jcm-12-00793-t005:** Mid-term postoperative surgical and functional outcomes and complications of studies on Trans Perineal Laser Ablation (TPLA) of prostate included in this review.

Study	12-mo IPSS	12-mo QoL	12-mo Qmax	12-mo PVR	12-mo PV	12-mo IIEF-5	12-mo MSHQ-EjD 3 item	12-mo MSHQ-EjD Bother	12-mo PSA (ng/mL)	Overall Complication Rate (n; %)	Early (<30 Days) Complication Rate	Complication Description	Complication Treatment	Clavien Dindo (n; %)
Sessa et al. [12]	Not reported	Not reported	Not reported	Not reported	Not reported	Not reported	Not reported	Not reported	Not reported	0; 0	0; 0	-	-	-
Cai et al. [13]	Not reported	Not reported	Not reported	Not reported	Not reported	Not reported	Not reported	Not reported	Not reported	2; 10	2; 10	1 urethral burn; 1 transient urine retention	1 catheter retained for 25 days; 1 catheter retained for 28 days	Not reported
De Rienzo et al. [14]	Not reported	Not reported	Not reported	Not reported	Not reported	Not reported	Not reported	Not reported	Not reported	1; 4.7	1; 4.7	1 prostatic abscess	1 percutaneous drainage and antibiotics	III (4.7)
Frego et al. [15]	6 (4.25–7)	1 (1–2)	20.5 (14.25–23.75)	30 (5–50)	41.5 (36.25–55)	21.5 (17.25–23.75)	Not reported	Not reported	Not reported	13; 59	12; 54.5	8 dysuria; 3 acute urine retention; 2 urinary tract infections	3 catheter retained 7 more days; 2 hospitalization and antibiotics	I (36.3); II (22.7)
Manenti et al. [16]	6.2 ± 3.8	2.1 ± 1.1	16.2 ± 4.9	18.8 ± 8.5	48.12 ± 19.2	22 ± 3	7.7 ± 3.2	Not reported	Not reported	3; 6.8	3; 6.8	1 prolonged haematuria; 1 orchitis; 1 bilateral prostatic abscess	1 medical treatment for orchitis; 1 percutaneous drainage and antibiotics	I (2.3); II (2.3); III (2.3)
Pacella et al. [17]	7.0 ± 2.9	1.6 ± 0.9	15.0 ± 4.0	17.8 ± 51.0	58.8 ± 22.9	Not reported	Not reported	Not reported	Not reported	8; 4.9	8; 4.9	3 transient hematuria 3 acute urinary retention; 1 orchitis; 1 prostatic abscess	3 catheter retained 15 more days; 1 percutaneous drainage	I (4.3); III (0.6)
Patelli et al. [18]	Not reported	Not reported	Not reported	Not reported	Not reported	Not reported	Not reported	Not reported	Not reported	0; 0	0; 0	-	-	-

Data presented as mean ± SD or as median (interquartile range). Mo: months; IPSS: International Prostate Symptom Score; QoL: Quality of Life; Qmax: Maximum flow rate; PVR: Post Void Residual; PV: Prostate Volume; IIEF-5: International Index of Erectile Function, five items; MSHQ-EjD: Male Sexual Health Questionnaire—Ejaculatory Disfunction; PSA: Prostate Specific Antigen.

## Data Availability

The data presented in this study are available on request from the corresponding author.

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
