# Peer review of "Transperineal Laser Ablation of the Prostate (TPLA) for Lower Urinary Tract Symptoms Due to Benign Prostatic Obstruction"

_jcm, 2023, doi:10.3390/jcm12030793_

Round 1

Reviewer 1 Report

It's a well written review on currently available minimal invasive TPLA for BPO data, which is an interesting upcoming technique.

In order to improve the manuscript it is advised:

- to write a systematic review to create a more scientifically robust review. This would improve search strategy yield (for example use of Embase and include conference article data of current trials), structured outcome analysis, heterogeneity and risk of bias analysis.

- to include other laser systems performing transperineal laser ablation for BPO. 

- to create a graphical illustration of most important outcomes, such as IPSS, IIEF, uroflowmetry; for readability. 

- to perform English spell check. 

Reviewer 2 Report

The authors have done a comprehensive review regarding TPLA. I have went through their manuscript with interest, without finding any readability issues  

Despite the fact that this is an interesting article, its methodology is weak. I would restructure it as a systematic review, adding a flow chart and other things that are missing in its current form.

Although an interesting article, it doesn't add anything to current literature around TPLA. 

Round 2

Reviewer 2 Report

An improved manuscript.